# MSGSA: Multi-Scale Guided Self-Attention Network for Crowd Counting

Yange Sun [1,2,*], Meng Li [1], Huaping Guo [1,2,*] and Li Zhang [1]

1    School of Computer and Information Technology, Xinyang Normal University, Xinyang 464000, China; limeng1997@xynu.edu.cn (M.L.); zhangli@xynu.edu.cn (L.Z.)
2    Research Center of Precision Sensing and Control, Institute of Automation, Chinese Academy of Sciences, Beijing 100190, China
*    Correspondence: yangesun@xynu.edu.cn (Y.S.); hpguo@xynu.edu.cn (H.G.)

**Abstract:** The use of convolutional neural networks (CNN) for crowd counting has made significant progress in recent years; however, effectively addressing the scale variation and complex backgrounds remain challenging tasks. To address these challenges, we propose a novel Multi-Scale Guided Self-Attention (MSGSA) network that utilizes self-attention mechanisms to capture multi-scale contextual information for crowd counting. The MSGSA network consists of three key modules: a Feature Pyramid Module (FPM), a Scale Self-Attention Module (SSAM), and a Scale-aware Feature Fusion (SFA). By integrating self-attention mechanisms at multiple scales, our proposed method captures both global and local contextual information, leading to an improvement in the accuracy of crowd counting. We conducted extensive experiments on multiple benchmark datasets, and the results demonstrate that our method outperforms most existing methods in terms of counting accuracy and the quality of the generated density map. Our proposed MSGSA network provides a promising direction for efficient and accurate crowd counting in complex backgrounds.

**Keywords:** crowd counting; self-attention; convolutional neural networks; multi-scale feature

---





## 1. Introduction

Crowd counting is a critical task in computer vision that involves estimating the number, density, or distribution of individuals in crowded scenes from images or videos [1–3]. As a subcategory of object counting, it has received growing attention due to its potential applications in various critical areas, including public safety management [4,5], traffic monitoring [6], and emergency management [7]; however, precisely estimating the number of people in crowds is a challenging task, primarily due to the existence of scale variation, and complex backgrounds [8,9]. To address these challenges, a variety of computer vision techniques are employed, including traditional methods [10–12] and deep learning-based approaches, such as CNN [13–21]. It is necessary to develop more robust and efficient models for crowd counting in real-world scenarios.

CNN is a commonly used deep architecture in crowd counting tasks, which involves generating density maps from crowd images and subsequently estimating the total number of individuals in the image [3]. Song et al. [22] proposed P2PNet, a pixel-based framework that employs normalized density average precision instead of mean absolute error. Cheng et al. [23] developed D2CNet, a two-stage decoupled framework that includes probability map regression and count map regression. Liu et al. [24] employed encoding–decoding structures and region attention modules to adjust head size in different positions; however, CNN-based methods generate density maps that solely reflect local relationships between short-distance pixels, neglecting global information between long-distance pixels, which could compromise counting accuracy in complex crowd counting scenarios.

Recent studies have explored the utilization of the Transformer model in crowd counting [25–28]. The Transformer is a deep learning model commonly used for sequence

modeling, which leverages the self-attention mechanism to capture long-range dependencies within the sequence. These studies have shown that the Transformer can effectively capture global information between distant pixels in complementing the local feature extraction performed by CNN [29–32]; however, a significant limitation of the Transformer model is its fixed scale of the token sequence, which restricts the self-attention mechanism to only consider interdependencies among tokens at the same scale [33–35]. This constraint limits the construction of a multi-scale feature architecture that is essential for improving the accuracy of crowd counting [36–38]. Compared to single-scale features, multi-scale feature fusion offers several advantages in crowd counting. Multi-scale features can better handle complex and cluttered backgrounds in crowd images by combining information from different scales. Figure 1 illustrates two intuitively plausible solutions for addressing the fixed-scale limitation of the Transformer, involving direct modification of the token sequence length or indirect construction of a multi-scale token sequence through image scaling. Liu et al. [39] proposed a down-sampling operation suitable for transformers, similar to the first solution; however, the second solution is more convenient to implement, as it only involves scaling the original image to construct a multi-scale token sequence.

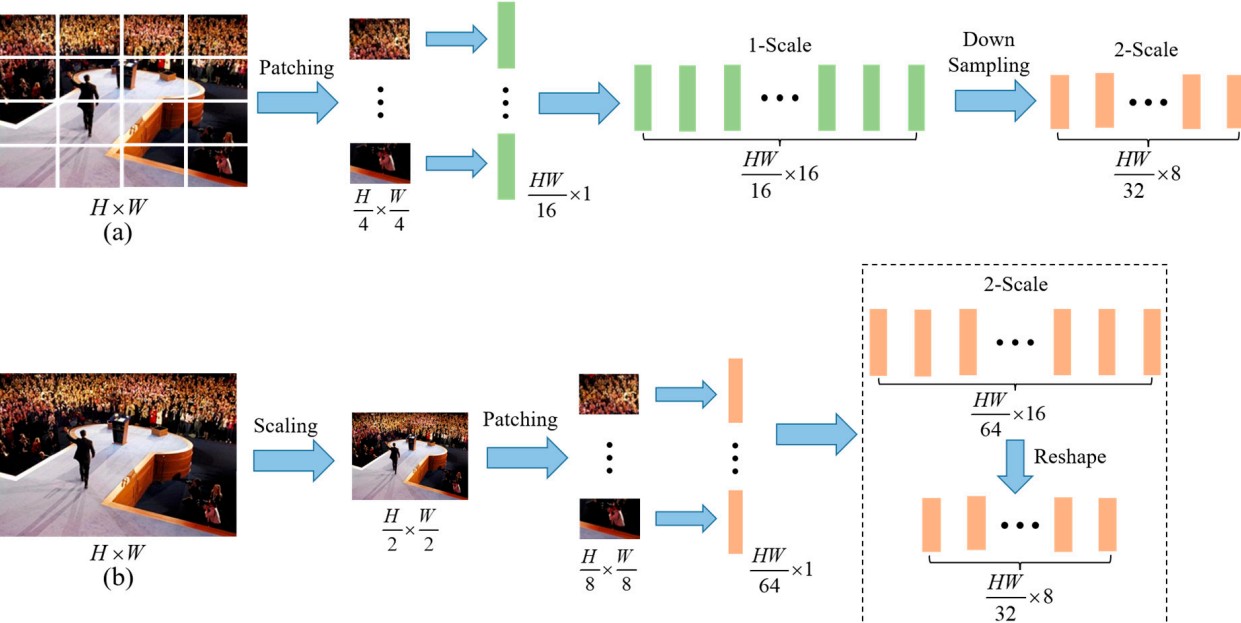

**Figure 1.** A comparison of two methods for reducing token sequence length. We employ a sliding window of size $4 \times 4$ and stride 4 to partition an image of size $H \times W$. In (**a**), the crowd image is directly partitioned into token sequences and down-sampled to obtain token sequences of different scales. In (**b**), the token sequence length is indirectly modified by resizing the crowd images to various scales.

To the best of our knowledge, no crowd counting method currently employs this approach; therefore, we introduce a novel network named Multi-Scale Guided Self-Attention (MSGSA), which integrates multi-scale feature representation and self-attention mechanism to enhance the accuracy and robustness of crowd counting. The MSGSA network consists of three key modules: a Feature Pyramid Module (FPM), a Scale Self-Attention Module (SSAM), and a Scale-aware Feature Aggregation (SFA). The FPM is utilized to extract multi-scale features from crowd images in order to achieve a multi-scale structure in the Transformer. The SSAM utilizes the Transformer to capture global contextual information of the multi-scale feature, facilitating the interaction between global information and local features. The SFA further aggregates the global and local information of multi-scale features, and generates accurate density maps that reflect the spatial distribution of crowds. The main contributions of this study are summarized as follows.

(1) We design the FPM and the SSAM, which facilitate the interaction between local features and global semantic information. As a result, our approach effectively mitigates the impact of scale variations and complex backgrounds in crowd images.

(2) We introduce the SFA that combines features from different scales to create a scale-aware feature representation. By integrating the global and local information of multi-scale features, the SFA generates accurate density maps that reflect the spatial distribution of crowds.

(3) The proposed method achieved remarkable performance on various benchmark datasets, surpassing most existing methods in terms of accuracy and quality of generated density maps. This state-of-the-art performance demonstrates the effectiveness and potential of our approach in crowd counting.

The structure of this paper is as follows. In Section 2, we provide a comprehensive review of the related literature. In Section 3, we describe the motivation and basic intuition behind our approach, and present the details of our methodology. In Section 4, we present the results of several comparison experiments and ablation studies. Finally, in Section 5, we summarize our findings and suggest directions for future research.

## 2. Related Work

### 2.1. Traditional Crowd Counting

Two main categories of traditional crowd counting methods can be identified: detection-based methods and regression-based methods. Detection-based methods aim to identify every pedestrian within the scene; however, traditional detection-based methods have limitations, as they struggle to accurately identify individuals in highly congested and occluded environments. Sliding window approaches, such as that proposed by Dolla et al. [10], can work well for sparse crowds, but they suffer from significant drawbacks in practical scenarios. Felzenszealb et al. [11] proposed a classifier that identifies pedestrians using partial body features, but this approach still exhibits significant errors in highly crowded scenes. To overcome these limitations, Chen et al. [12] introduced an adaptive regression model that trains a mapping between the image features and the count of individuals; however, the above methods rely on manually extracted features to identify pedestrians, making them susceptible to scale changes and background occlusion, which reduces their prediction accuracy in practical applications.

### 2.2. Crowd Counting Based on CNN

CNN-based methods have emerged as a powerful alternative to traditional approaches, offering improved performance and greater flexibility. To tackle issues such as scale variation and complex background, researchers have explored two key avenues, namely multi-scale structures [13–15] and attention mechanisms [16–18]. One prominent example is the multi-column convolutional neural network (MCNN) proposed by Zhang et al. [19], which utilizes multiple convolutional kernels of varying sizes to extract features with different receptive fields. Similarly, Sam et al. [20] proposed a density classification network (switch-CNN) that adaptively outputs density levels through a density classifier. Yang et al. [21] proposed a perspective-inverted network that tackles the issue of scale variation by estimating the perspective coefficient and distorting the image.

In the task of crowd counting, attention mechanisms have been extensively utilized to tackle occlusion challenges and enhance counting accuracy. One common approach is to incorporate attention mechanisms in the feature extraction process. Li et al. [40] used cross-modal recurrent attention fusion to combine RGB and depth features to model the crowd distribution. Shi et al. [13] introduced a multi-scale spatial attention perception network, which utilizes dilated convolution to extract multi-scale feature maps and then applies spatial attention to these maps to address the issue of scale variation. Gu et al. [41] designed a context-aware pyramid feature extraction module that uses multi-level contextual information to improve the counting performance. These CNN-based methods have demonstrated promising results, achieving high accuracy in crowd counting tasks;

however, due to the limited receptive field of CNN convolutional kernels, CNNs do not fully integrate the global information of crowd images. Recent studies have explored crowd counting methods based on Transformers to effectively combine the global information of crowd images [25–28].

### 2.3. Crowd Counting Based on Transformer

The Transformer's self-attention mechanism has shown promise as an alternative to traditional CNN-based methods for computer vision tasks, including crowd counting. Unlike CNNs that depend on manually designed convolutional kernel sizes, the Transformer's self-attention mechanism enables the model to dynamically concentrate on the most relevant parts of the input data from a global perspective, leading to effective feature representation and learning. Liang et al. [25] used weak supervision to train TransCrowd, which utilizes the self-attention mechanism of Transformer and shows promise for crowd counting. CrowdFormer, proposed by Yang et al. [26], implements a multi-scale structure of Transformer using overlapping convolutions. Deng et al. [27] proposed a semi-supervised model that combines Transformer and CNN, including multi-level convolutional Transformers and adaptive scaling modules. Lin et al. [28] proposed a multi-modal attention network that uses local attention mechanism to dynamically allocate attention to each feature position. Compared to CNN-based methods, Transformer-based models exhibit significant advantages in handling complex backgrounds and dense crowd datasets, reflecting the rapid development and promising future of Transformer-based architectures.

## 3. Methodology

In this section, we present a comprehensive overview of our method. It comprises three modules: the Feature Pyramid Module (FPM), the Scale Self-Attention Module (SSAM), and the Scale-aware Feature Aggregation (SFA).

### 3.1. Network Architecture

The task of crowd counting is confronted with significant challenges arising from scale variation and complex backgrounds in crowd images. Existing methods, employing CNN to address this issue, often focus solely on the local features of crowd images, overlooking the importance of global information. Transformer applies self-attention mechanism to capture global information between long-distance pixels, but it still has the issue of fixed scale; therefore, aiming to balance the importance of local features and global information in identifying scale variation and complex background, we propose MSGSA. The proposed network architecture comprises three stages, illustrated in Figure 2. Firstly, CNN is employed as the backbone of the FPM to extract multi-scale features from the input crowd images. Secondly, the SSAM module utilizes self-attention to enhance the model's representation capability by capturing relevant information from multi-scale features extracted by the FPM and achieve information interaction between global and local features. Finally, the SFA module combines the multi-scale features to obtain scale-aware features and generates high-quality crowd density maps using multiple convolutional layers, resulting in a robust and efficient crowd density estimation framework.

### 3.2. Feature Pyramid Module

In our framework, the FPM is designed to generate multi-scale features by scaling the original image and extracting local features and semantic information through convolutional neural networks. The proposed approach allows for the capturing of information at multiple levels of detail. Different CNN architectures can be used as the FPM in MSGSA to extract multi-scale local features. As shown in Figure 2, to present the structure of MSGSA more intuitively, we depict the MSGSA network architecture using VGG16 as the FPM module. We utilize VGG16 with 5 convolutional blocks and corresponding pooling layers that progressively decrease the resolution of the crowd image to 1/2, 1/4, 1/8, 1/16, and 1/32. The deep convolutional features contain rich semantic information, while the shallow

features focus more on detail features such as edges, textures, and contours. By retaining the feature outputs from all five convolutional blocks, we obtain multi-scale features that serve as the input for the Transformer. This approach assists in resolving the challenges of fixed scale and inadequate local feature representation in Transformer. Specifically, we remove the pooling layer in the final convolutional block to ensure that the feature maps retain sufficient spatial information at larger scales.

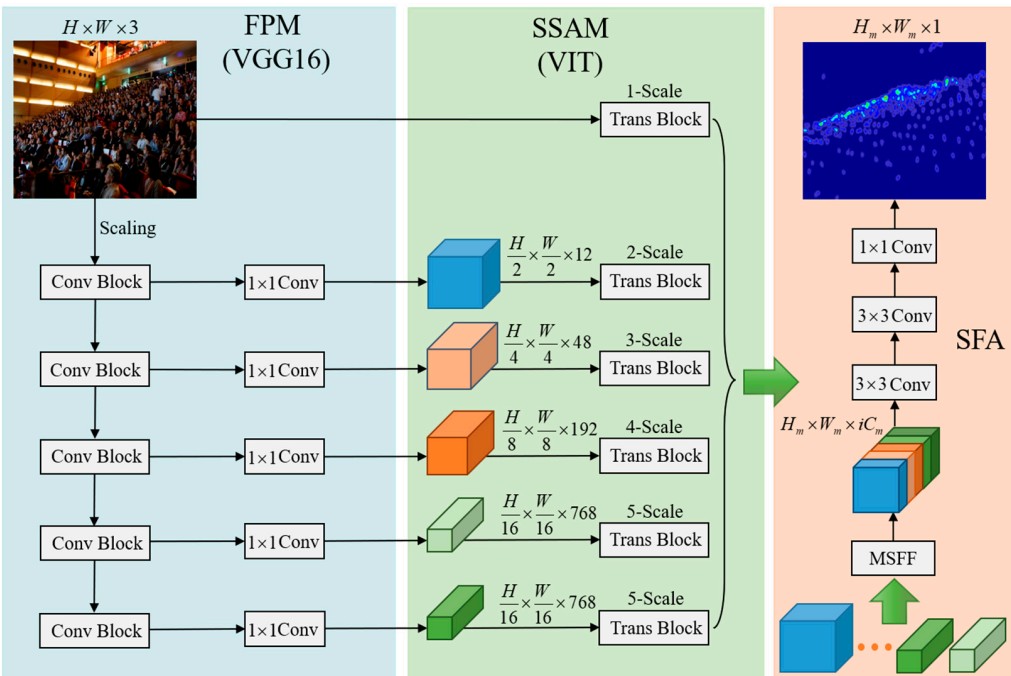

**Figure 2.** The architecture of the MSGSA.

As the depth of the network increases, the feature map resolution decreases progressively, and the number of channels increases significantly. To avoid computational and memory overhead for the Transformer's self-attention mechanism, we add a $1 \times 1$ convolution to each convolutional block to adjust the channel of each feature map, while maintaining a consistent total number of pixels in each feature map. It is important to note that when the patching size of the Transformer is fixed, using multi-scale features as input indirectly alters the length of the token sequence. This allows each token to capture more spatial local information, enhancing the inductive bias of the Transformer. It can be defined as follows:

$$x_i = C(x_0) \tag{1}$$

where $x_0 \in \mathbb{R}^{H \times W \times 3}$ is the given crowd image, and $x_i \in \mathbb{R}^{\frac{H}{2^i} \times \frac{W}{2^i} \times C_i}$ denotes the multi-scale features extracted by the CNN.

### 3.3. Scale Self-Attention Module

When observing crowd images, humans can easily identify the scale variation and complex background in a crowd image through global and local contrasts; however, the multi-scale features extracted by FPM lack interaction with global crowd information, although rich in spatial local information and semantic information. We adopt a vision transformer (VIT) block to model human visual mechanisms and implement the multi-scale structure of the Transformer. By feeding the multi-scale features extracted by FPM, we utilize the multi-head self-attention mechanism to capture global contextual information and promote interaction between global and local features. VIT divides the original image into patches of size $16 \times 16$, which can only obtain self-attention at a coarse granularity. In contrast, we divide the multi-scale feature map into patches of size $4 \times 4$, thus obtaining

self-attention at a fine granularity. We also employ a Transformer block with a depth of 2 to extract self-attention across features at various scales, effectively controlling the model's complexity.

In Figure 3, the Transformer Block converts the multi-scale feature map $x_i$ into a 2D token sequence $x_i' \in \mathbb{R}^{N_i \times D_i}$ by applying $K \times K$ sliding window with stride K. Then, we use the multi-head attention mechanism to map $x_i'$ to a multi-dimensional feature subspace, which captures the global contextual information and facilitates the interaction between local features and global information. It can be defined as follows:

$$f_i = MSA(x_i') + x_i' \tag{2}$$

$$F_i = project(f_i, w_i) \tag{3}$$

where MSA stands for multi-head self-attention, $project(*)$ maps the patch sequence to a standard image form $F_i \in \mathbb{R}^{\frac{H}{2^i} \times \frac{W}{2^i} \times C_i}$, and $w_i$ is a learnable position parameter, which ensures that the relative position of each patch in the image remains unchanged.

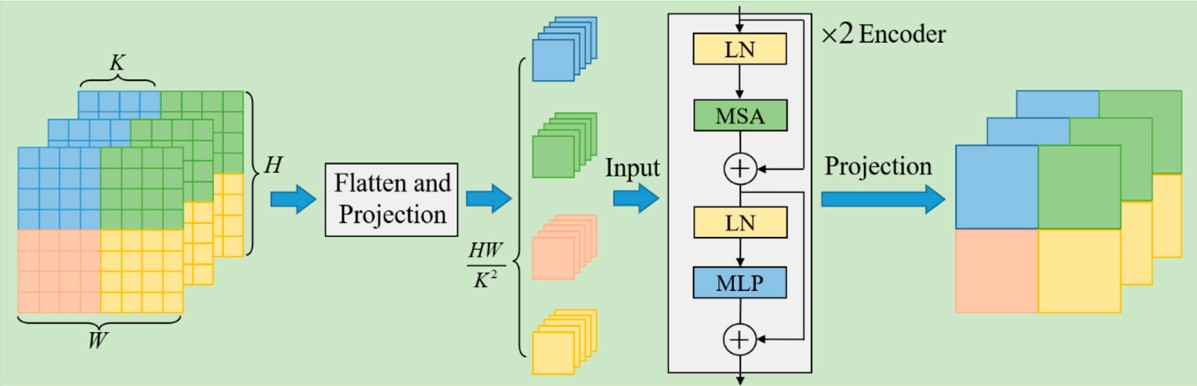

**Figure 3.** The architecture of proposed Transformer block.

### 3.4. Scale-Aware Feature Aggregation Module

We have designed the SFA module to enable information exchange between multiple scales and generate crowd density maps, as shown in Figure 4. The module first adjusts the channel number of the multi-scale features through $1 \times 1$ convolution, reducing computational cost. Then, global average pooling compresses the spatial information of each channel, while the sigmoid function activates each channel to obtain its weight vector. Finally, the multi-scale features are up-sampled to a specified resolution $H_m \times W_m$ and concatenated along the channel dimension to generate scale-aware features $y'_0 \in \mathbb{R}^{H_m \times W_m \times (i+1)C_m}$. These features are further aggregated using two $3 \times 3$ convolutional layers, and the crowd density map is generated using a final convolutional layer. The definition of this part is as follows:

$$y'_0 = cat(SE(F_0), \cdots, SE(F_p) \cdots, SE(F_i)) \tag{4}$$

$$y_0 = C'(y'_0) \tag{5}$$

where $y_0 \in \mathbb{R}^{H_m \times W_m \times 1}$ represents the density map of the crowd image $x_0$.

### 3.5. Loss Functions

The loss function employed by our method utilizes the Euclidean distance during training to supervise the error between the predicted and real density maps. Specifically, it is defined as follows:

$$L_2(\theta) = \frac{1}{N} \sum_{i=1}^{N} \left\| y_i^{pred}(x_i, \theta) - y_i^{truth} \right\|^2 \tag{6}$$

where $N$ is the number of training samples, $y_i^{pred}(x_i, \theta)$ represents the predicted density map of input image $x_i$, and $y_i^{truth}$ represents the real density map.

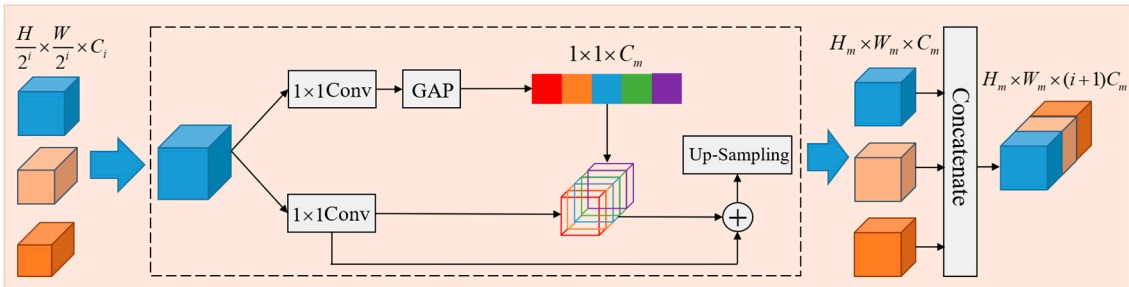

**Figure 4.** The architecture of multi-scale feature fusion.

## 4. Experiments and Discussion

We extensively evaluate our proposed method on three datasets using four 3090TI GPUs implemented in Pytorch. We use the Adam optimizer with a batch size of four, learning rate of $1 \times 10^{-5}$, and weight decay of $1 \times 10^{-4}$. Furthermore, we conduct thorough ablation experiments to validate the effectiveness of each component of our proposed network.

### 4.1. Datasets

We evaluate the performance of our proposed method using three benchmark datasets, as summarized in Table 1. The NWPU-Crowd dataset has the largest number of annotations and covers a wide range of crowd sizes, while the UCF-QNRF dataset has the highest average number of people per image.

**Table 1.** Comparison with different datasets.

| Dataset | Scale | Number | Annotation | Max | Mean | Min |
|---|---|---|---|---|---|---|
| ShanghaiTech Part_A | unfixed | 482 | 241,677 | 3139 | 501 | 33 |
| ShanghaiTech Part_B | $1024 \times 768$ | 716 | 88,488 | 578 | 124 | 9 |
| UCF-QNRF | unfixed | 1535 | 1,251,642 | 12,865 | 815 | 49 |
| NWPU-Crowd | unfixed | 5109 | 2,133,375 | 20,033 | 418 | 0 |

The ShanghaiTech dataset contains 1198 crowd images with a total of 330,165 annotated heads, divided into two parts: Part_A and Part_B. Part_A includes 482 internet-collected crowd images, with 300 designated for training and 182 for testing, while Part_B consists of 716 surveillance-captured crowd images, with 400 for training and 316 for testing. Part_A has a higher crowd density compared to Part_B, which features a sparse crowd distribution.

The UCF-QNRF dataset comprises 1535 high-resolution internet-sourced images with 1,251,642 annotated heads, divided into 1201 training and 334 test images. This dataset presents a significant challenge due to its dense crowd distribution and substantial scale variations.

NWPU-Crowd is the most extensive dataset with 5109 crowd images and 2,133,375 annotated heads, including 3109 training images, 1500 test images, and 500 validation images. Among them, 3109 images constitute the training set, 500 form the validation set, and 1500 form the test set. It offers several advantages, including large data volume, higher resolution, and significant variation.

### 4.2. Performance Metrics

To evaluate the counting accuracy and robustness of our proposed method, we adopt two widely used evaluation metrics: mean absolute error (MAE) and mean squared error

(MSE). MAE represents the average absolute difference between the predicted and ground-truth crowd counts, while MSE indicates the average squared difference between them. The definitions of these metrics are as follows:

$$\text{MAE} = \frac{1}{N}\sum_{i=1}^{N}\left|y_i^{\text{gt}} - y_i^{\text{pred}}\right| \tag{7}$$

$$\text{MSE} = \sqrt{\frac{1}{N}\sum_{i=1}^{N}\left(y_i^{\text{gt}} - y_i^{\text{pred}}\right)^2} \tag{8}$$

where $N$ denotes the total number of test images, and $y_i^{pred}$ and $y_i^{gt}$ represent the predicted result and signifies the ground truth of the *i*-th image, respectively.

### 4.3. Comparison with Different Backbones

We conducted evaluations of our model using various CNNs as FPM, and the corresponding results are summarized in Table 2. ResNet-101 achieved the best performance across all datasets, improving MAE and MSE by 1.5 and 3.5 on Part_A, 0.6 and 0.8 on Part_B, 1.9 and 11.1 on NWPU-Crowd, 0.9 and 3.5 on UCF-QNRF compared to VGG16. Its advantage lies in the incorporation of the residual structure in ResNet-101, which optimizes the model's feature extraction capability, enhances the spatial local information of multi-scale feature maps, and facilitates better global and local interaction, providing an advantage in crowd datasets with complex backgrounds and drastic scale changes.

**Table 2.** Comparison with different backbones.

| Backbone | Part_A | | Part_B | | NWPU-Crowd | | UCF-QNRF | |
|---|---|---|---|---|---|---|---|---|
| | MAE | MSE | MAE | MSE | MAE | MSE | MAE | MSE |
| VGG16 | 59.3 | 94.8 | 7.5 | 11.9 | 75.8 | 334.6 | 81.7 | 141.0 |
| Resnet-50 | 58.6 | 92.9 | 7.2 | 11.4 | 74.4 | 327.3 | 80.9 | 139.8 |
| Resnet-101 | 57.8 | 91.3 | 6.9 | 10.8 | 73.9 | 323.5 | 80.8 | 137.5 |

Compared to ResNet-50, ResNet-101 improved MAE and MSE by 0.8 and 1.6 on Part_A, 0.3 and 0.6 on Part_B, 0.5 and 3.8 on NWPU-CROWD, 0.1 and 2.3 on UCF-QNRF, owing to its deeper residual structure that can extract more local features and higher-level semantic information.

### 4.4. Comparison with the State-of-the-Art

In this section, we conducted extensive experiments on ShanghaiTech, NWPU-Crowd, and UCF-QNRF, to evaluate the proposed model's performance. The results presented in Table 3 indicate that our method outperforms most existing methods in terms of counting accuracy and density map quality, achieving state-of-the-art performance.

MSGSA demonstrated notable enhancements on the NWPU-Crowd and UCF-QNRF datasets, indicating its robust performance in challenging and complex crowd scenes. Specifically, the model showed improvements of 0.6 and 3.9 in MAE and MSE, respectively, on NWPU-Crowd. Similarly, it achieved improvements of 0.4 and 1.1 on UCF-QNRF. These results demonstrate the model's robustness and practicality in crowd counting tasks. The proposed method's superior performance is due to multi-scale feature extraction, self-attention aggregation, and effective interaction between local and global information, improving counting accuracy and robustness in ultradense crowd scenes. Our proposed method makes a substantial contribution to crowd counting by integrating feature representation and self-attention mechanism to enhance accuracy and robustness. By incorporating Transformers to capture global information, our approach effectively integrates global and local information from multiple scales to generate highly accurate density maps that depict the spatial distribution of crowds. The aggregation of features from various scales leads to

improved feature representation and adaptation to different crowd scenes, enhancing the overall robustness and generalization ability of the model.

**Table 3.** Comparison with state-of-the-art methods.

| Method | Year | Part_A | | Part_B | | NWPU-Crowd | | UCF-QNRF | |
|---|---|---|---|---|---|---|---|---|---|
| | | MAE | MSE | MAE | MSE | MAE | MSE | MAE | MSE |
| TransCrowd [25] | 2022 | 66.1 | 105.1 | 9.3 | 16.1 | 117.7 | 451.0 | 97.2 | 168.5 |
| NoiseCC [42] | 2020 | 61.9 | 99.6 | 7.4 | 11.3 | 96.9 | 534.2 | 85.8 | 150.6 |
| Gloss [43] | 2021 | 61.3 | 95.4 | 7.3 | 11.4 | 79.3 | 346.1 | 84.3 | 147.5 |
| KDMG [44] | 2020 | 63.8 | 99.2 | 7.8 | 12.7 | 100.5 | 415.5 | 99.5 | 173.0 |
| DM-count [45] | 2020 | 59.7 | 95.7 | 7.4 | 11.3 | 88.4 | 357.6 | 85.6 | 148.3 |
| BM-count [46] | 2021 | 57.3 | 90.7 | 7.4 | 11.8 | 83.4 | 358.4 | 81.2 | 138.6 |
| UOT [47] | 2021 | 58.1 | 95.9 | 6.5 | 10.2 | 87.8 | 387.5 | 83.3 | 142.3 |
| CA-Net [48] | 2019 | 62.3 | 100.0 | 7.8 | 12.2 | -- | -- | 107.0 | 183.0 |
| DKPNet [18] | 2021 | 55.6 | 91.0 | 6.6 | 10.9 | 74.5 | 327.4 | 81.4 | 147.2 |
| P2PNet [22] | 2021 | 52.7 | 85.1 | 6.2 | 9.9 | 77.4 | 362.0 | 85.3 | 154.5 |
| SASNet [49] | 2021 | 53.6 | 88.4 | 6.4 | 9.9 | -- | -- | 85.2 | 147.3 |
| Ours | -- | 57.8 | 91.3 | 6.9 | 10.8 | 73.9 | 323.5 | 80.8 | 137.5 |

The performance of MSGSA on the ShanghaiTech is not satisfactory. On one hand, this could be due to the fact that ShanghaiTech contains fewer and lower-resolution crowd images compared to NWPU-Crowd and UCF-QNRF. This results in a smaller amount of training data for the Transformer model, limiting its ability to fully capture crowd features. On the other hand, it could be attributed to the differences in data distribution between the training and testing sets.

We further provide insight into the exceptional performance of our approach by visualizing some density maps. As demonstrated by the red markers in Figure 5, MSGSA effectively addresses crowd scenes characterized by significant scale variation, demonstrating outstanding proficiency in extracting comprehensive global contextual information across multiple scales. Additionally, our proposed method exhibits strong performance in generating accurate density maps for dense crowd scenes, effectively addressing the challenges presented by complex backgrounds and scale variations through the integration of the FPM, SSAM, and SFA modules. This is demonstrated by the high consistency between our generated density maps and ground-truth annotations, indicating the efficacy of our approach in achieving precise crowd density estimations.

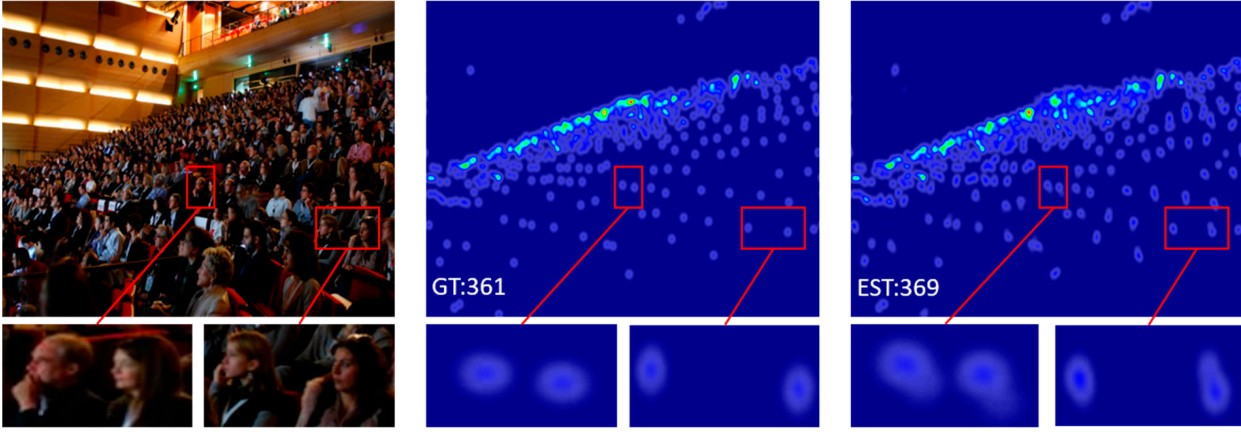

**Figure 5.** Crowd scenes with continuous scale variation.

Furthermore, our approach demonstrates a remarkable ability to remove the interference of intricate backgrounds (such as banners, flags, etc.) in crowded scenes, as highlighted by the red markers in Figure 6. This is attributed to the effective functioning of SSAM and SFA, which facilitate the model's ability to capture interdependencies between different scales of input features, leading to precise crowd region segmentation from complex scenes.

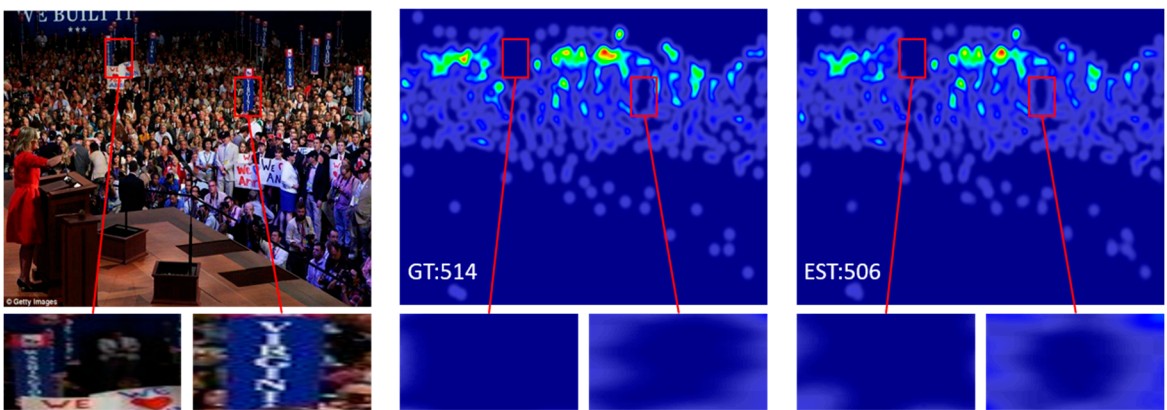

**Figure 6.** Crowd scenes with complex background.

*4.5. Ablation Studies*

We perform a series of ablation experiments on ShanghaiTech Part A to assess the contribution of each module in our proposed method. Specifically, we employ VGG16 as the FPM in our implementation, and incrementally incorporate the corresponding modules to validate their efficacy.

**Effect of SSAM.** As illustrated in Table 4, when comparing the FPM with FPM + SSAM, the latter demonstrates superior performance, improving MAE by 7.6 and MSE by 15.9. We conducted a visualization of the density maps generated by FPM and FPM + SSAM to validate the effectiveness of SSAM. The results, as shown by the red markers in Figure 7, demonstrate that incorporating SSAM enhances the model's sensitivity to high-density crowds, resulting in the generation of more accurate density maps that precisely reflect the spatial distribution of people compared to using FPM alone.

**Table 4.** Performance of different modules in MSGSA.

| Module | ShanghaiTech Part_A | |
| :---: | :---: | :---: |
| | **MAE** | **MSE** |
| FPM | 69.1 | 114.3 |
| FPM + SSAM | 61.5 | 98.4 |
| FPM + SSAM | 59.3 | 94.8 |

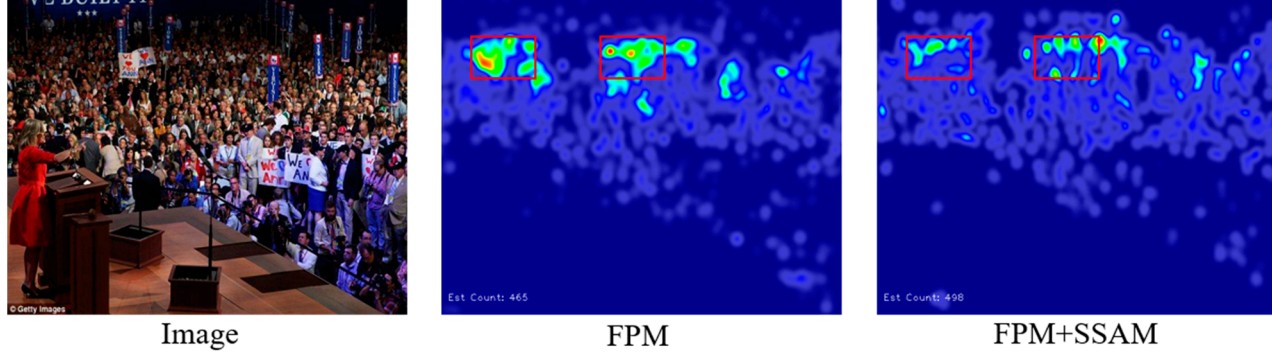

**Figure 7.** Comparison of visualization results between FPM and FPM + SSAM.

This improvement can be attributed to the organic fusion of FPM and SSAM. On the one hand, the FPM module enables the model to capture information at different levels of detail, which is particularly useful for crowd counting where individuals can vary in size and density. By extracting features at multiple scales, the FPM can better handle variations in object size and position, as well as the complex and cluttered backgrounds often present in crowd images. The resulting multi-scale features are then fed into subsequent modules for further processing and refinement.

The SSAM module, on the other hand, employs the self-attention mechanism to capture global semantic information across various scales. By considering the relationships between all pixels in the image, SSAM effectively capture the underlying spatial structure and patterns in the crowd image, leading to improved feature representation and overall model performance. Furthermore, SSAM uses the Transformer's self-attention mechanism to facilitate interaction between local and global features, enhancing the model's adaptability to complex scenes and improving its accuracy and robustness.

**Effect of SFA**. Table 4 illustrates that the performance is significantly improved by incorporating SSAM and SFA into the FPM. Specifically, compared to the FPM, the FPM + MSSA + MSFF enhances the MAE by 2.2 and the MSE by 3.6. To further verify the effectiveness of SFA, we conducted a visualization of the density maps generated by FPM + SSAM and FPM + SSAM + SFA.

The red markers in Figure 8 show that the addition of SFA to FPM + SSAM effectively removes redundant information and improves the model's ability to predict high-density crowd areas, resulting in more accurate crowd density maps.

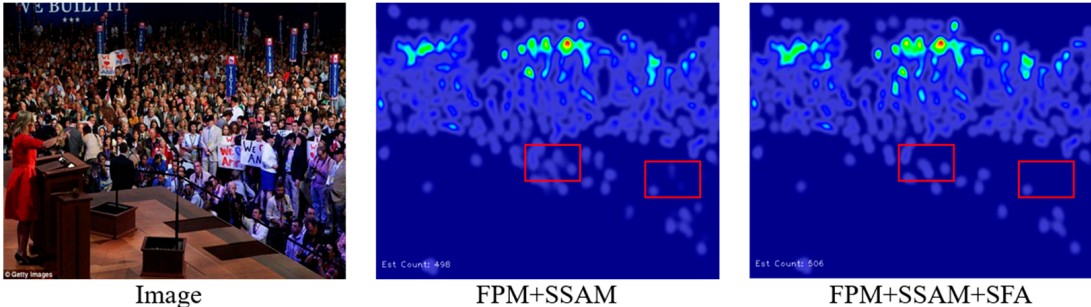

**Figure 8.** Comparison of visualization results between FPM + SSAM and FPM + SSAM + SFA.

The success of MSFF can be attributed to its capability to aggregate features from multiple scales and assign varying attention weights to different channels, which helps the network to better focus on crowd regions and enhance the overall performance.

## 5. Conclusions

In this paper, we introduce a novel approach called Multi-Scale Guided Self-Attention (MSGSA) network for accurate and efficient crowd counting in complex backgrounds with scale variation. By utilizing self-attention mechanisms at different scales, our MSGSA network captures multi-scale contextual information, thereby enhancing the accuracy of crowd counting. The combination of Transformer and CNN allows the MSGSA network to achieve precise crowd density estimations. The study includes a comprehensive ablation study and visualization of density maps to analyze the contributions of each module of the MSGSA network. The results demonstrate the effectiveness of the MSGSA network in dealing with dense crowd scenes with complex backgrounds.

Although the proposed method has shown promising results for efficient and accurate crowd counting in complex backgrounds, there is still potential for future research and improvements. Future work will focus on investigating the MSGSA network's robustness in real-world scenarios by testing it on more diverse datasets with varying lighting conditions and weather patterns. Additionally, we plan to incorporate domain adaptation techniques

to improve the network's generalization to novel and unobserved situations, addressing the issue of domain shift between training and testing datasets.

**Author Contributions:** Y.S.: methodology, original draft, writing; M.L.: formal analysis, resources, writing; H.G.: review and editing, Visualization; L.Z.: review and editing, supervision. All authors have read and agreed to the published version of the manuscript.

**Funding:** This research was funded by the National Natural Science Foundation of China (62062004), the Natural Science Foundation of Henan Province (222300420274, 222300420275, 232300421167), the Key Scientific Research Projects of Henan Province (22A520008, 22A220002), the Academic Degrees & Graduate Education Reform Project of Henan Province (22A520008, 22A220002),the Postgraduate Education Reform and Quality Improvement Project of Henan Province (YJS2023SZ23), and the Nanhu Scholars Program for Young Scholars of XYNU.

**Data Availability Statement:** The datasets generated during and/or analyzed during the current study are available from the corresponding author upon reasonable request. The Shanghai_Tech dataset can be downloaded from https://svip-lab.github.io/dataset/campus_dataset.html; the UCF_QNRF dataset can be downloaded from https://www.crcv.ucf.edu/data/ucf-qnrf/; and the NWPU_Crowd dataset can be downloaded from https://gjy3035.github.io/NWPU-Crowd-Sample-Code/.

**Conflicts of Interest:** The authors declare no conflict of interest.

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
