# Peer review of "MSGSA: Multi-Scale Guided Self-Attention Network for Crowd Counting"

_electronics, doi:10.3390/electronics12122631_

Round 1
Reviewer 1 Report
In this paper, the authors present "MSGSA: Multi-Scale Guided Self-Attention Network for Crowd Counting." The authors conducted extensive experiments on multiple benchmark datasets, and the results demonstrate that the method outperforms most existing methods in terms of counting accuracy and the quality of the generated density map. The proposed MSGSA network provides a promising direction for efficient and accurate crowd counting in complex backgrounds. However, there are some issues should be addressed.
1. Precisely estimating the number of people in crowds is a challenging task, primarily due to the existence of scale variation, and complex backgrounds. How to solve this problem in this paper?
2. CNN-based methods generate density maps that solely reflect local relationships between short-distance pixels, neglecting global information between long-distance pixels, which could compromise counting accuracy in complex crowd counting scenarios. How to solve this problem in this paper?
3. A significant limitation of the Transformer model is its fixed scale of the token sequence, which restricts the self-attention mechanism to only consider inter-dependencies among tokens at the same scale. How to solve this problem in this paper?
4. Do the methods proposed in this paper have limitations, for example in identify individuals in highly congested and occluded environments?
5.Does the method proposed in this paper rely on manually extracted features to identify pedestrians? Is the method proposed in this paper sustainable to scale changes and background occlusion, which reduces the prediction accuracy in practical applications?
6.Are the features extracted in this paper lack interaction with global crowd information?
Extensive editing of English language required
Author Response
Dear Prof.,
On behalf of my co-authors, we would express our sincere thanks for. We also greatly appreciate the constructive comments and suggestions made by the reviewers and/or editors on our manuscript entitled “MSGSA: Multi-Scale Guided Self-Attention Network for Crowd Counting” (ID: electronics-2419963).
We have read the reviewer’s comments and suggestions carefully and have made significant revision as required accordingly. We are uploading (a) our point-by-point response to the comments (below) (response to reviewers), (b) an updated manuscript with yellow highlighting indicating changes, and (c) a clean updated manuscript without highlights (PDF main document).
Looking forward to the publication of this manuscript shortly with your satisfaction.
Thanks, with best regards.
Yours sincerely,
Yange Sun
Corresponding author: Yange Sun, Huaping Guo
Email: yangesun@xynu.edu.cn (Y.S.); hpguo@xynu.edu.cn (H.G.)
- Response to the Reviewer#1’s Comments
---------------------------------------the Reviewer #1’s Comments--------------------------------------------
The authors have carried out some of my suggestions, but there are still some points to be clarified:
In this paper, the authors present "MSGSA: Multi-Scale Guided Self-Attention Network for Crowd Counting." The authors conducted extensive experiments on multiple benchmark datasets, and the results demonstrate that the method outperforms most existing methods in terms of counting accuracy and the quality of the generated density map. The proposed MSGSA network provides a promising direction for efficient and accurate crowd counting in complex backgrounds. However, there are some issues should be addressed.
- Precisely estimating the number of people in crowds is a challenging task, primarily due to the existence of scale variation, and complex backgrounds. How to solve this problem in this paper?
- CNN-based methods generate density maps that solely reflect local relationships between short-distance pixels, neglecting global information between long-distance pixels, which could compromise counting accuracy in complex crowd counting scenarios. How to solve this problem in this paper?
- A significant limitation of the Transformer model is its fixed scale of the token sequence, which restricts the self-attention mechanism to only consider inter-dependencies among tokens at the same scale. How to solve this problem in this paper?
- Do the methods proposed in this paper have limitations, for example in identify individuals in highly congested and occluded environments?
- Does the method proposed in this paper rely on manually extracted features to identify pedestrians? Is the method proposed in this paper sustainable to scale changes and background occlusion, which reduces the prediction accuracy in practical applications?
- Are the features extracted in this paper lack interaction with global crowd information?
------------------------------------------------------------------------------------------------------------------------
Thank you for your valuable comments and suggestions.
- Response to Comment (1):
Thanks for the comment. The solution to the problem of scale variation and complex background issues in academic writing involves the following two aspects. On one hand, we model the human visual mechanism by enabling the network to perceive scale variation and complex backgrounds in crowd images through the interaction of global information (Transformer) and local features (CNN).The corresponding content of the article can be found in Section 3.1, lines 171-173. On the other hand, we construct a multi-scale Transformer structure that allows the network to extract self-attention representations at different granularities, thereby enhancing the network's ability to filter out complex backgrounds. The corresponding content of the article can be found in Section 3.1, lines 175-177.
- Response to Comment (2):
Thanks for the comment. As you mentioned, the CNN-based methods partially overlook the global information in crowd images. Therefore, in our approach for crowd counting tasks, we incorporate the Transformer model, leveraging its self-attention mechanism to fully extract the global information of crowd images. The corresponding content of the article can be found in Section 1, lines 79-81.
- Response to Comment (3)
Thanks for the comment. We utilize CNN to extract feature maps at multiple scales and feed these feature maps into the Transformer to address the issue of fixed-scale token sequences. The corresponding content of the article can be found in Section 1, lines 77-79.
- Response to Comment (4)
Thanks for the comment. As you mentioned, while our approach demonstrates excellent counting performance on multiple complex datasets, there is still significant room for improvement in counting performance when faced with extremely dense crowd scenes. In future work, we will further discuss and investigate this aspect.
- Response to Comment (5)
Thanks for the comment. The method proposed in this paper does not rely on manually extracted features for pedestrian recognition, but instead utilizes deep features obtained by neural networks to identify pedestrians. The proposed approach benefits from the organic combination of CNN and Transformer, which models the efficient visual contrast mechanism of humans and enhances the predictive accuracy in practical applications.
- Response to Comment (6)
Thanks for the comment. Indeed, when using only CNN as a feature extractor, there is a limitation in capturing the global crowd information. However, our method addresses this issue by combining CNN with Transformer, thereby fully considering the global information in crowd images.

Reviewer 2 Report
The motivation for using the proposed deep learning topology could be explained in some more details. Similarly some discussion for the cases where the system does not perform well. This will explain the justification of proposed extenssions for future work.
Author Response
Dear Prof.,
On behalf of my co-authors, we would express our sincere thanks for. We also greatly appreciate the constructive comments and suggestions made by the reviewers and/or editors on our manuscript entitled “MSGSA: Multi-Scale Guided Self-Attention Network for Crowd Counting” (ID: electronics-2419963).
We have read the reviewer’s comments and suggestions carefully and have made significant revision as required accordingly. We are uploading (a) our point-by-point response to the comments (below) (response to reviewers), (b) an updated manuscript with yellow highlighting indicating changes, and (c) a clean updated manuscript without highlights (PDF main document).
Looking forward to the publication of this manuscript shortly with your satisfaction.
Thanks, with best regards.
Yours sincerely,
Yange Sun
Corresponding author: Yange Sun, Huaping Guo
Email: yangesun@xynu.edu.cn (Y.S.); hpguo@xynu.edu.cn (H.G.)
- Responds to the Reviewer #2’s Comments
---------------------------------------the Reviewer#2’s Comments--------------------------------------------
The motivation for using the proposed deep learning topology could be explained in some more details. Similarly some discussion for the cases where the system does not perform well. This will explain the justification of proposed extensions for future work.
------------------------------------------------------------------------------------------------------------------------
Thank you for your valuable comments and suggestions. According to your suggestion, we updated the manuscript by:
- We have rewritten the introduction, to provide a better description. And we have redrawn Figure 1 in Section 1 to provide a clearer illustration of our motivation. The modification is marked in yellow highlighting.
(2) We have rewritten the Section 3 to give a clearer and better explanation of our method. The modification is marked in yellow highlighting.
(3) We have also supplemented the analysis of cases where the system does not perform well at Section 4.4. The modification is marked in yellow highlighting.

Reviewer 3 Report
Review
The submitted paper entitled: “MSGSA: Multi-Scale Guided Self-Attention Network for 2 Crowd Counting” presents a new approach for crowd counting based on self-attention. The main idea is very interesting.
The paper is very well written.
Following are my remarks and proposition to improve the paper:
Authors need to add a brief description on the idea behind the use of Pyramid Module (FPM) and Scale-aware Feature Fusion 15 (SFA).
In the introduction, to define the crowd counting you used the reference [1] which not accurate. This reference also cites other papers when defining it.
Check the reference [4] used for the use of crowd counting in safety management is not correct. And, the DOI used.
Figure 1 is not clearly readable and not of good quality.
Check the formatting of the 6th paragraph of the first section; as follows: and the numerical list
The section “related work” is very informative.
In the section 3.1, the following sentence is not correct:
“Generating density maps for crowd images has been challenging due to scale variation and complex backgrounds. Existing methods using CNN to address this issue have not fully considered global information.”
In the proposed approach section, you need to justify the choice of the VGG16? Is this choice crucial or you can use any other DL architecture? And what is bizarre is that in section 4-3 you demonstrate that ResNet-101 is the best in features extraction.
In the whole paper you were talking about self-attention, but in the Methodology section you introduced the VIT (Vision transformer). This is not acceptable. You should notice from the beginning that you will use the VIT. VIT is general concept that uses self-attention.
You propose to use a 4x4 patches instead of 16x16 without justifying your choice.
In table 3, some approaches references are missing.
Author Response
Dear Prof.,
On behalf of my co-authors, we would express our sincere thanks for. We also greatly appreciate the constructive comments and suggestions made by the reviewers and/or editors on our manuscript entitled “MSGSA: Multi-Scale Guided Self-Attention Network for Crowd Counting” (ID: electronics-2419963).
We have read the reviewer’s comments and suggestions carefully and have made significant revision as required accordingly. We are uploading (a) our point-by-point response to the comments (below) (response to reviewers), (b) an updated manuscript with yellow highlighting indicating changes, and (c) a clean updated manuscript without highlights (PDF main document).
Looking forward to the publication of this manuscript shortly with your satisfaction.
Thanks, with best regards.
Yours sincerely,
Yange Sun
Corresponding author: Yange Sun, Huaping Guo
Email: yangesun@xynu.edu.cn (Y.S.); hpguo@xynu.edu.cn (H.G.)
- Responds to the Reviewer#3’s Comments
---------------------------------------the Reviewer #3’s Comments--------------------------------------------
The submitted paper entitled: “MSGSA: Multi-Scale Guided Self-Attention Network for 2 Crowd Counting” presents a new approach for crowd counting based on self-attention. The main idea is very interesting. The paper is very well written.
Following are my remarks and proposition to improve the paper:
- Authors need to add a brief description on the idea behind the use of Pyramid Module (FPM) and Scale-aware Feature Fusion 15 (SFA).
- In the introduction, to define the crowd counting you used the reference [1] which not accurate. This reference also cites other papers when defining it.
- Check the reference [4] used for the use of crowd counting in safety management is not correct. And, the DOI used.
- Figure 1 is not clearly readable and not of good quality.
- Check the formatting of the 6th paragraph of the first section; as follows: and the numerical list
- The section “related work” is very informative.
- In the section 3.1, the following sentence is not correct:
“Generating density maps for crowd images has been challenging due to scale variation and complex backgrounds. Existing methods using CNN to address this issue have not fully considered global information.”
- In the proposed approach section, you need to justify the choice of the VGG16? Is this choice crucial or you can use any other DL architecture? And what is bizarre is that in section 4-3 you demonstrate that ResNet-101 is the best in features extraction.
- In the whole paper you were talking about self-attention, but in the Methodology section you introduced the VIT (Vision transformer). This is not acceptable. You should notice from the beginning that you will use the VIT. VIT is general concept that uses self-attention.
- You propose to use a 4x4 patches instead of 16x16 without justifying your choice.
- In table 3, some approaches references are missing.
------------------------------------------------------------------------------------------------------------------------
Thank you for your valuable comments and suggestions.
- Response to Comment (1):
Thank you very much for providing valuable suggestions. We have incorporated the relevant descriptions into the sixth paragraph of Section 1 (line 76-line 84), and the corresponding modifications have been highlighted in yellow.
- Response to Comment (2):
Thanks for the comment. We have replaced Reference [1] with a review article that defines crowd counting. The corresponding modifications in the reference section have been highlighted in yellow.
- Response to Comment (3)
Thanks for the comment. We have reviewed and replaced Reference [4], and the corresponding modifications in the reference section have been highlighted in yellow.
- Response to Comment (4)
Thank you for your valuable suggestions. We have redrawn Figure 1 and provided a detailed description. The corresponding modifications in the introduction section have been highlighted in yellow.
- Response to Comment (5)
Thank you for your valuable suggestions. We have made modifications to the formatting of Section 1 (line 85- line 87 ). This modification is marked in yellow highlighting.
- Response to Comment (6)
Thank you very much for your suggestions. We have made revisions to the relevant work and removed unnecessary content. This modification is marked in yellow highlighting.
- Response to Comment (7)
Thank you for your valuable suggestions. The corresponding changes have been highlighted in yellow at the Section 3.1 (line 166-line 169).
- Response to Comment (8)
This was an error in our paper writing. Our intention was to convey that MSGSA, as a crowd counting framework, can utilize various CNN architectures (VGG16, ResNet) as the FPM module to extract multi-scale local features. Introducing VGG16 would provide a more intuitive presentation of MSGSA's network structure. In order to avoid ambiguity and misunderstanding, we have provided a description in Section 3.2 (line 188-line 190), and the corresponding modifications have been highlighted in yellow.
- Response to Comment (9)
Thank you for your valuable suggestions. According to your suggestion, we have taken note of this issue and have been extremely cautious while making modifications, particularly with regards to the usage of Transformers and self-attention mechanisms. This modification is marked in yellow highlighting (Section 1 line 79 - line 81).
- Response to Comment (10)
Thank you for your valuable suggestions. We have added the corresponding explanations in Section 3.3 (line 220 - line 222), and the relevant modifications have been highlighted in yellow.
- Response to Comment (11)
Thank you for your valuable suggestions. We have carefully reviewed Table 3 to ensure the accuracy of the references cited in each article.

Round 2
Reviewer 1 Report
The authors have solved the related problems. It is good enough.
Minor editing of English language required